

# A systematic review and realist synthesis on toilet paper hoarding: COVID or not COVID, that is the question

Javier Labad[1,2,3], Alexandre González-Rodríguez[3,4,5], Jesus Cobo[2,3,4,5], Joaquim Puntí[5] and Josep Maria Farré[6]

[1] Department of Mental Health, Consorci Sanitari del Maresme, Mataró, Spain
[2] Centro de Investigación Biomédica en Salud Mental (CIBERSAM), Madrid, Spain
[3] Institut d'Investigació Sanitària Parc Taulí (I3PT), Sabadell, Spain
[4] Department of Psychiatry and Legal Medicine, Universitat Autónoma de Barcelona, Cerdanyola del Vallès, Spain
[5] Department of Mental Health, Parc Taulí Hospital Universitari, Sabadell, Spain
[6] Department of Psychiatry, Psychology and Psychosomatics, Dexeus University Hospital, Barcelona, Spain

Corresponding author
Javier Labad, jlabad@csdm.cat

## ABSTRACT

**Objective:** To explore whether the coronavirus disease 2019 (COVID-19) pandemic is associated with toilet paper hoarding and to assess which risk factors are associated with the risk of toilet paper hoarding.
**Design:** A systematic review and realist review were conducted.
**Data sources:** PubMed, Web of Science, Scopus and PsycINFO were searched (systematic review). PubMed, pre-prints and grey literature were also searched (realist review). The databases were searched from inception until October 2020.
**Study selection:** There were no restrictions on the study design.
**Outcomes and measures:** For the systematic review, toilet paper hoarding was the main outcome, and pathological use of toilet paper was the secondary outcome. For the realist review, the context-mechanisms-outcome (CMO) scheme included the COVID-19 pandemic (context), four proposed mechanisms, and one outcome (toilet paper hoarding). The four potential mechanisms were (1) gastrointestinal mechanisms of COVID-19 (e.g. diarrhoea), (2) social cognitive biases, (3) stress-related factors (mental illnesses, personality traits) and (4) cultural aspects (e.g. differences between countries).
**Eligibility criteria for selecting studies:** All studies of human populations were considered (including general population studies and clinical studies of patients suffering from mental health problems).
**Results:** The systematic review identified 14 studies (eight studies for the main outcome, six studies for the secondary outcome). Three surveys identified the role of the COVID-19 threat in toilet paper hoarding in the general population. One study pointed to an association between a personality trait (conscientiousness) and toilet paper buying and stockpiling as well as an additional significant indirect effect of emotionality through the perceived threat of COVID-19 on toilet paper buying and stockpiling. Six case reports of pathological use of toilet paper were also identified, although none of them were associated with the COVID-19 pandemic. The realist review suggested that of all the mechanisms, social cognitive biases and a bandwagon effect were potential contributors to toilet paper hoarding in the general population.

The stressful situation (COVID-19 pandemic) and some personality traits (conscientiousness) were found to be associated with toilet paper hoarding. Cultural differences were also identified, with relatively substantial effects of toilet paper hoarding in several Asian regions (Australia, Japan, Taiwan and Singapore).

**Conclusions:** The COVID-19 pandemic has been associated with a worldwide increase in toilet paper hoarding. Social media and social cognitive biases are major contributors and might explain some differences in toilet paper hoarding between countries. Other mental health-related factors, such as the stressful situation of the COVID-19 pandemic, fear of contagion, or particular personality traits (conscientiousness), are likely to be involved.

**Registration:** PROSPERO CRD42020182308

## INTRODUCTION

Toilet paper, sometimes called toilet tissue or loo roll, is defined by the Merriam-Webster dictionary as 'a thin sanitary absorbent paper usually in a roll for use in drying or cleaning oneself after defecation and urination.' To wipe is human, and paper has been used for faecal-related cleaning purposes since the end of the 6th century in China, although the toilet paper industry blossomed in the early 14th century in China during the reign of the Yang dynasty (*Smyth, 2012*). The commercial use of toilet paper started in 1857 thanks to Joseph Gayetty, a New York-based entrepreneur who sold medicated paper infused with aloe that aimed to cure haemorrhoids (*Smyth, 2012*). The reception of toilet paper from the medical community was not positive, and in an ironic note published in the Lancet in 1869, the idea of toilet paper was defined as 'the last absurdity' (*The Lancet, 1869*). The note was sarcastic when referring to the opinion of Gayetty, who anticipated that 'this article will be found in the household of every refined man in the kingdom.' Many years later, toilet paper has become an essential product for a great proportion of the population worldwide.

Since early December 2019, the coronavirus SARS-CoV-2 has spread from Wuhan (China) to many countries worldwide, causing the coronavirus disease 2019 (COVID-19). In terms of deaths, COVID-19 has been the worst pandemic since the 1918 flu pandemic, also known as the Spanish flu (although its origin was in Kansas, USA (*Worobey, Cox & Gill, 2019*)). By 11 November 2020, the COVID-19 pandemic had caused at least 51,251,715 infections and 1,270,930 deaths (*WHO, 2021*). During the first months of the pandemic, medical masks were in short supply in most countries. This issue was expected because SARS-CoV-2 is viable and infectious in aerosols for hours (*Van Doremalen et al., 2020*), and using face masks is advised in situations where meeting others is likely, as masks can reduce the risk of transmitting the infection (*Greenhalgh et al., 2020*). The global toilet paper shortage amid the coronavirus was much less expected, but in the weeks that followed the pandemic spread, compulsive panic buying of toilet paper was observed in many countries on different continents (*Buchholz, 2020*). Toilet paper became

a co-star with coronavirus in the news in many countries, with surprising information every week: rationing of toilet paper by supermarkets (*Pidd, 2020*), toilet rolls being chained to their dispensers in public toilets (*Lewis, 2020*), armed robbers stealing hundreds of paper rolls (*Leung, 2020*), and deserted supermarket and grocery shelves (*Knoll, 2020*). People were buying and hoarding toilet paper even before it was known that the virus could be detected in the faeces of infected patients (*Chen et al., 2020a*) or that approximately 10% of COVID-19 patients may suffer from diarrhoea (*Miri et al., 2020*). Therefore, a scientific question demands an urgent response: why do people hoard toilet paper?

We aimed to shed light on potential risk factors associated with toilet paper hoarding, with a particular interest in stress-related situations such as the COVID-19 pandemic. As hoarding is often seen in patients with obsessive-compulsive disorder (OCD) and other psychiatric disorders as well as in people with obsessive-compulsive traits (*Mataix-Cols et al., 2010*), it is important to make the differential diagnosis with mental health problems. In most individuals, compulsive hoarding appears to be a syndrome distinct from OCD, which is associated with substantial levels of disability and social isolation (*Pertusa et al., 2008*). This has led to the inclusion of hoarding as a separate diagnosis in the 5th edition of the Diagnostic and Statistical Manual of Mental Disorders (DSM-5). One hoarding criterion is the acquisition of and failure to discard a large number of possessions that seem to be useless or of limited value (*Mataix-Cols et al., 2010*). We are not currently interested in addressing the debate about the utility of toilet paper, but it is important to mention that to date, toilet paper is not a specifier of the DSM-5 diagnostic criteria for hoarding. Moreover, there is no previous evidence suggesting that toilet paper hoarding is a behaviour distinct from other hoarding behaviour. Epidemiological studies suggest that hoarders are older, often unmarried, and more likely to be impaired by a current physical health condition or comorbid mental disorder (*Nordsletten et al., 2013*). There is limited information regarding the prevalence of toilet paper hoarding in the general population.

It is also important to underscore that compulsive buying and hoarding are two related phenomena, as hoarding is a predictor of compulsive buying (*Lawrence, Ciorciari & Kyrios, 2014*). Moreover, people with buying-shopping disorder report more hoarding symptoms than healthy control individuals (*Vogel et al., 2019*). Both buying and hoarding behaviour have been described as being preceded by stressful life events and traumatic experiences (*Tolin et al., 2010*; *Landau et al., 2011*; *Vogel et al., 2019*). For this reason, it is important to study how stress influences hoarding behaviour because this knowledge would help to understand some of the recent panic-buying behaviour seen in the weeks following the COVID-19 pandemic.

As hoarding behaviours are observed in both non-clinical (*Bulli et al., 2014*) and clinical (*Pertusa et al., 2008*) samples, studies considering non-clinical populations need to be considered. It is also important to analyse whether the mechanisms linking stress with toilet paper hoarding are shared by people with mental disorders (hoarding disorders and other psychiatric disorders) and the general population or whether this relationship might change depending upon the social or cultural context.

| Context | Mechanisms | Outcome |
|---------|-----------|---------|
| COVID-19 pandemic | M1. COVID-19 disease is associated with diarrhoea (or polyuria), which contributes to the panic buying and toilet paper hoarding | Toilet paper hoarding |
| | M2. Social cognitive biases and social media are facilitators of toilet paper hoarding in the general population | |
| | M3. The COVID-19 pandemic is a stressful event that causes the exacerbation of mental illnesses and hoarding behaviours leading to toilet paper hoarding | |
| | M4. Cultural aspects moderate the relationship between the COVID-19 pandemic and toilet paper hoarding, with differences between countries | |

**Table 1 CMO scheme of the realist review.**

Neuroimaging studies using functional magnetic resonance imaging (fMRI) and conducting experimental approaches (provocation of hoarding-related anxiety) in healthy subjects and clinical populations (OCD patients) have demonstrated that hoarding symptoms are associated with activation of the same brain areas, involving the anterior ventromedial prefrontal cortex, in both non-clinical and clinical populations (*Mataix-Cols et al., 2003*; *An et al., 2009*). These results support a common transdiagnostic neurobiological pathway for hoarding symptoms. Interestingly, the anterior ventromedial prefrontal cortex has also been implicated in buying behaviour in studies exploring value-based decisions with a buying task during an fMRI session (*Gluth, Rieskamp & Büchel, 2012*).

The main objective of our study was to identify potential mechanisms linking the context of a stressful situation (COVID-19 pandemic) with a specific outcome (toilet paper hoarding). We also aimed to study potential mechanisms involved in hoarding behaviour that might be influenced by psychopathological, psychological, social and cultural determinants that could act as moderators.

To achieve these objectives, we conducted one study that included two sequential steps:

First, we conducted a systematic review exploring potential risk factors associated with toilet paper hoarding. Psychopathology, personality and stress-related factors (including pandemics, especially the COVID-19 pandemic) were considered. The main hypothesis of our systematic review was that a substantial proportion of the general population would hoard toilet paper amid the COVID-19 pandemic. As a secondary aim of the systematic review, we also wanted to study whether toilet paper use (pathological use and/or hoarding) is associated with negative mental health outcomes (e.g. greater risk of depression, suicide). This secondary aim was exploratory in nature.

Second, we conducted a realist review exploring different theory-driven mechanisms on potential moderators of the relationship between the COVID-19 pandemic and toilet paper hoarding (Table 1). A realist review is based on a realist philosophy of science and considers the interaction among context, mechanism and outcome, also known as the CMO configuration (*Wong et al., 2013*). As explained in the RAMESES guidelines for realist syntheses (*Wong et al., 2013*), this type of review uses the concept of a mechanism for understanding the relationship between context and outcome. Several mechanisms might be studied and can be defined as 'underlying entities, processes, or (social) structures which operate in particular contexts to generate outcomes of interest.' A realist review is

an interpretative type of literature review, in contrast with a systematic review that attempts to collect all empirical evidence that fits pre-specified eligibility criteria in order to answer a specific research question (*Berg & Nanavati, 2016*). Regarding the realist review, four hypotheses were formulated in relation to different mechanisms that might partially explain toilet paper hoarding during the COVID-19 pandemic: (1) diarrhoea or polyuria contributes to panic buying and toilet paper hoarding; (2) social cognitive biases and social media facilitate the hoarding of toilet paper; (3) stress contributes to the worsening of mental health outcomes that could also increase the risk of toilet paper hoarding; and (4) cultural aspects moderate the relationship between the COVID-19 pandemic and toilet paper hoarding.

Finally, several recommendations for future research will be included considering the gaps in the scientific literature. Clinical and ecological implications of our research will also be summarised.

## MATERIALS AND METHODS

### Systematic review

Preferred Reporting Items for Systematic Reviews and Meta-analyses (PRISMA) guidelines (*Moher et al., 2009*) were followed. The protocol was registered in PROSPERO (CRD42020182308).

#### Search strategy

Four electronic bibliographic databases were searched: PubMed, Web of Science, Scopus and PsycINFO. The following search strategy was used: (Toilet AND (paper OR tissue OR roll)) AND (psychiatry OR psychology OR mental OR anxiety OR depression OR schizophrenia OR bipolar OR psychosis OR delusion OR personality OR neuroticism OR obsessive OR hoarding OR suicide OR stress* OR pandemic OR epidemic OR COVID-19 OR coronavirus OR virus). The search strategy was performed by J.L. and A.G.R. Studies published through 31 October 2020 were considered for inclusion.

#### Inclusion and exclusion criteria

In our systematic review, toilet paper hoarding was considered the main outcome. This outcome was defined as a behavioural pattern characterised by excessive acquisition of and an inability or unwillingness to discard large quantities of toilet paper that cause significant distress or impairment. This definition is in agreement with the current DSM-5 diagnostic category for hoarding disorder, but it has been adapted for specifying that the main item saved is toilet paper.

We also conducted a secondary analysis for the systematic review that considers toilet paper (pathological use or hoarding) as a risk factor for mental health outcomes (depression, suicide, etc.).

In those studies using toilet paper as an outcome (e.g. toilet paper hoarding), all potential exposures (stress-related situations, personality factors, psychopathology, and mental illnesses) were considered. In those studies with toilet paper use as an exposure, the considered outcomes were mental health problems (e.g. depression, suicide).

All types of studies that relate to mental health or stress-related aspects of toilet paper use were included. There were no restrictions on the types of study design. All studies conducted in human populations (general population studies and clinical studies of patients suffering from mental health problems) were considered for inclusion. Language was restricted to articles written English, Spanish, Catalan, Portuguese, Dutch, French, or German. There was no restriction on the type of document indexed in the electronic databases (these documents could include original articles, reviews, letters to the editor, case reports, editorials, conference proceedings). We included published documents in the systematic review, and therefore, preprints were not included in the systematic review (they could be included in the realist review).

### Data collection and extraction

All retrieved records were checked for duplicates using Covidence (https://www.covidence.org/). The titles and/or abstracts of studies retrieved using the search strategy and those from additional sources were screened independently by two review authors (J.L. and A.G.R.) to identify studies that met the inclusion criteria. Any disagreement between them over the eligibility of particular studies was resolved through discussion with two additional reviewers (J.C. and J.P). The flow chart of all selected studies is described in Fig. 1.

### Risk of bias (quality) assessment

Quality assessment was conducted with the Newcastle Ottawa Scale (cohort and case–control studies) (Wells et al., 2012) or the CARE guidelines (case reports) (Riley et al., 2017). Case reports and case series are also rated with the tool for evaluating the methodological quality of case reports and case series (Murad et al., 2018).

## Realist review

A realist synthesis was conducted following the RAMESES guidelines (Wong et al., 2013). An additional reviewer (J.C.) participated in the search for potential citations along with the two researchers participating in the systematic review (J.L. and A.G.R.). We started by considering all reviewed items in the previous step with the theory-driven approach of the realist review. Four mechanisms were tested (Table 1). Iterative screening was completed by these reviewers, who also conducted additional searches to explore these hypotheses on PubMed as well as grey literature available on the internet (e.g. Google searching). Search terms differed for each mechanism: (1) Mechanism 1: covid AND (diarrhoea OR polyuria); (2) Mechanism 2: (stress OR covid OR pandemic) AND cognitive bias AND social; (3) Mechanism 3: (covid OR stress OR pandemic) AND hoarding; and (4) Mechanism 4: (toilet paper OR hoarding) AND (culture OR cultural). All potential abstracts were included if they could contribute to explaining any of the four studied mechanisms linking the COVID-19 pandemic with toilet paper hoarding. The identification and selection of citations were guided by these research questions and were based on the trustworthiness of sources. This last characteristic is not easy to verify, as fake news is mixed with real news all over the internet. We tried to reduce
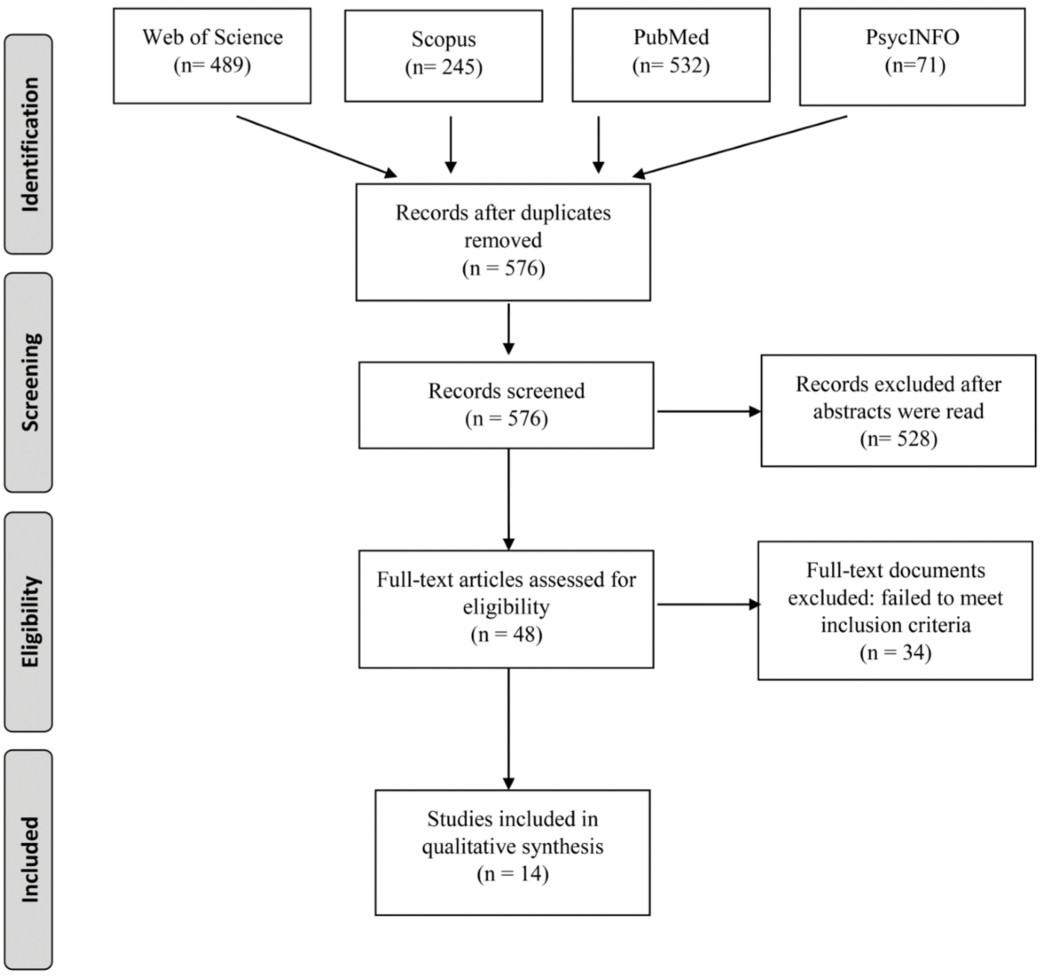

**Figure 1 PRISMA flow diagram of the studies included in the systematic review.**

the inclusion of fake news by carefully observing the sources, particularly when they came from non-peer-reviewed sources.

To explore differences in cultural aspects of toilet paper hoarding, we also verified the Google search trending topics during the year of 2020 in the world (https://trends.google.com/). Previous research indicates that Google search data are useful in predicting near-future consumer behaviour (*Goel et al., 2010*). The search frequency on Google has also been proposed as a direct measure of investor attention (*Da, Engelberg & Gao, 2011*). Data regarding the use of toilet paper were graphed with Excel (Microsoft Corporation, USA) after downloading the .csv file. We also compared two search terms (toilet paper vs covid) to analyse the relative popularity of the term 'toilet paper' with respect to the 'COVID' term.

Qualitative studies and material were managed with the software QDA miner Lite version 2.0.7 (Provalis Research, Canada). Data regarding the potential contribution of the studied mechanisms were extracted. To identify key elements of importance to the success or failure of a mechanism in a certain context using a realist perspective,
information was gathered on the mechanism, the context and the actual 'working of the mechanism.' The strength of the evidence and the usefulness of the application of realist principles to available data were discussed.

## RESULTS

### Systematic review

A total of 1,337 records were identified in initial searches (Web of Science: 489; Scopus: 245, PubMed: 532; PsycINFO: 71). After duplications were removed, 576 records were screened after reading the title and abstract. Further details of the screening and selection processes can be found in Fig. 1. Finally, 14 studies were included because they were focused on toilet paper hoarding behaviour and met our selection criteria.

Primary outcome: toilet paper hoarding

We identified eight published studies related to the COVID-19 pandemic with toilet paper hoarding (*Garbe, Rau & Toppe, 2020*; *Kirk & Rifkin, 2020*; *Oosterhoff & Palmer, 2020*; *Sim et al., 2020*; *Miri et al., 2020*; *Micalizzi, Zambrotta & Bernstein, 2020*; *Laato et al., 2020*; *Güzel, 2020*). One study included a survey of participants focused on toilet paper shopping and stockpiling behaviours (*Garbe, Rau & Toppe, 2020*), one study conducted a survey of adolescents regarding several pandemic-related behaviours (including hoarding) (*Oosterhoff & Palmer, 2020*), and another study conducted a survey through Amazon's Mechanical Turk on stockpiling behaviour (including toilet paper) in response to COVID-19 (*Micalizzi, Zambrotta & Bernstein, 2020*). One study analysed purchasing behaviour during the COVID-19 pandemic (*Laato et al., 2020*). The other four published studies included theoretical discussions on consumption behaviours, including panic buying during the COVID-19 pandemic (*Kirk & Rifkin, 2020*; *Sim et al., 2020*; *Miri et al., 2020*; *Güzel, 2020*).

The first study (*Garbe, Rau & Toppe, 2020*) explored the relationship between personality traits based on the HEXACO model (Honesty-Humility, Emotionality, eXtraversion, Agreeableness, Conscientiousness and Openness to experience). This study was a survey that included a final sample of 996 adults from 22 countries. Participants were asked about their perception of the level of threat posed by COVID-19 and their toilet paper consumption behaviour (shopping frequency, shopping intensity, number of toilet paper rolls stocked in their household). Older participants shopped more frequently, bought more packages of toilet paper and had more toilet paper rolls in stock than younger participants. Participants residing in Europe shopped for toilet paper significantly more frequently than North American residents but had less toilet paper in stock. In this study, participants were asked about whether they stocked toilet paper more than usual, which could be considered an indirect measure of toilet paper hoarding. Of all participants in the survey, 17.2% of North Americans and 13.7% of Europeans reported stockpiling toilet paper. The perceived threat of COVID-19 was positively related to all three toilet paper variables (shopping frequency, shopping intensity and toilet paper stockpiling). The HEXACO model suggested that participants scoring high in conscientiousness (organisation, diligence, perfectionism, and prudence) shopped for toilet paper more and stocked more toilet paper. This study also included an additional analysis exploring the

indirect effect of emotionality (fearfulness, anxiety, dependence, and sentimentality) on toilet paper consumption. They found a significant indirect effect of emotionality through the perceived threat of COVID-19 on shopping intensity and the amount of stocked toilet paper rolls. In the quality assessment with the Newcastle Ottawa Scale for this study, we considered the perceived threat of COVID-19 as the main exposure and toilet paper behaviour as the main outcome (definition of cases). The quality assessment yielded two stars for selection (representativeness of cases, selection of controls), two stars for comparability and one star for the definition of the exposure. Therefore, this study obtained five of nine possible stars on the Newcastle Ottawa Scale.

Another survey of 770 adolescents in the United States explored the role of psychological factors in pandemic-related behaviours during the COVID-19 outbreak (social distancing, disinfecting, monitoring the news, hoarding supplies) (*Oosterhoff & Palmer, 2020*). In this study, attitudes about the greater severity of COVID-19 and greater self-interest values were associated with more hoarding, whereas greater social responsibility and social trust were associated with less hoarding. In the quality assessment with the Newcastle Ottawa Scale for this study, we considered attitudes about the severity of COVID-19 as the main exposure and hoarding behaviour as the main outcome (definition of cases). This study obtained five out of nine stars on the Newcastle Ottawa Scale (selection (two stars), comparability (two stars), definition of exposure (one star)).

One survey of 363 workers in the United States who were recruited from Amazon's Mechanical Turk inquired about the stockpiling of 13 items, as well as opinions on the COVID-19 pandemic and political affiliation (Democrat vs. Republican) (*Micalizzi, Zambrotta & Bernstein, 2020*). Of all 13 items, toilet paper was the item most commonly stockpiled (63.2%). However, more than half of the sample reported stockpiling other supplies, such as canned goods (59.2%), rice (57.4%), bottled water (57.0%), pasta (56.2%), bread (53%) and medicine (52.7%). When looking at predictors of increased stockpiling with multivariate analyses that were adjusted for education status, income, age and number of people living at home, higher stockpiling was observed among those who were more conservative, worried more about the pandemic, had more people in the home, and reported less social distancing. This study obtained four out of nine stars on the Newcastle Ottawa Scale (selection (one star), comparability (two stars), definition of exposure (one star)).

Five studies reflected on potential explanations for toilet paper hoarding amid the COVID-19 pandemic and justified this behaviour with different hypotheses, such as a reaction to a threat to product availability that increases the perceived need for the threatened object and makes consumers behave with an emotionally reactive response (*Kirk & Rifkin, 2020*). Other potential moderators included the conflict between the desire to maintain regular routines versus the uncertainty of limiting access to daily necessities by the pandemic, a coping response to stressful unmet situations or even a reaction to the loss of control of the future and social pressures to conform to similar behaviours (*Sim et al., 2020*). Another study testing potential contributors of toilet paper hoarding during the COVID-19 pandemic in a Finnish sample (*Laato et al., 2020*) proposed a structured model connecting exposure to online information sources to two

behavioural responses (unusual purchases and voluntary self-isolation). Exposure to online information increased health anxiety as measured by cyberchondria and consequently the intention to make unusual purchases and engage in voluntary isolation.

In another systematic review on gastrointestinal symptoms of COVID-19 that indicates the long persistence of COVID-19 in the gastrointestinal tract after primary treatment (*Miri et al., 2020*), the authors suggested that these findings could explain the coronavirus-related panic buying of toilet rolls.

Finally, another study was a personal reflection about the panic buying and hoarding of toilet paper during the COVID-19 pandemic from a psychodynamic perspective (*Güzel, 2020*).

Secondary outcome: toilet paper (pathological use or hoarding) and mental health outcomes

Six case reports of pathological use of toilet paper hoarding were identified. One study reported a case of a patient with therapy-resistant OCD who spent hours on the toilet with excessive anus wiping, using at least 10 rolls of toilet paper per day (*Klimke et al., 2016*). Interestingly, with only two applications of transcranial alternating current stimulation (tACS), the patient showed immediate improvement (using less than one toilet roll per day).

Two case reports indicated suicide by mechanical asphyxia using toilet paper: one patient suffering from schizophrenia (*Sauvageau & Yesovitch, 2006*) and another patient with borderline personality disorder (*Saint-Martin, Bouyssy & O'Byrne, 2007*). It is not always easy to distinguish suicide from homicide, and another study reported the case of a homicide by toilet paper smothering in a patient with Alzheimer's disease (*Saint-Martin, Lefrancq & Sauvageau, 2012*).

Two other case reports described patients with pica, a syndrome characterised by unusual craving for the ingestion of either edible or inedible substances, who ate toilet paper (*Chisholm & Martin, 1981*; *Fisher et al., 2014*). The diagnosis of this syndrome is a clinical challenge because this condition might be underreported and is sometimes diagnosed after studying medical complications such as iron deficiency and gastrointestinal bleeding (*Fisher et al., 2014*). In other cases, biochemical deficiencies need to be studied because hypozincaemia might play a role in the ingestion of toilet paper (*Chisholm & Martin, 1981*).

All six case reports are described in Table 2. The quality of the studies assessed with the CARE guidelines (Table 2) and the recommendations by *Murad et al. (2018)* (Table S1) was good. None of these case reports was related to the COVID-19 pandemic.

## Realist review

The PubMed search for the four theory-driven mechanisms (M1–M4, Table 1) included a total of 452 records (M1: 108; M2: 104; M3: 80; M4: 60). After the review by three authors, 92 records were selected. Fourteen additional records from grey literature were also included.

COVID-19 disease is associated with diarrhoea (or polyuria), which contributes to panic buying and toilet paper hoarding (Mechanism #1)

**Table 2 Main characteristics of case reports included in the systematic review (*n* = 6).**

| Case | Author, year of publication | Age (y.o) | Gender | Substance use | Comorbid psychiatric diagnosis | Treatment | Primary outcome (toilet paper) | Secondary outcomes | Checklist CARE guidelines | |
|---|---|---|---|---|---|---|---|---|---|---|
| | | | | | | | | | Completed items | Missing sub-items[*][#] |
| 1 | Klimke et al. (2016) | 17 | Man | NR | OCD | tACS, lorazepam 0.5 mg day, | Before treatment: use of 10 rolls of toilet paper After treatment (two stimulations): one roll of toilet paper | None | 8/13 | 5c, 5d, 6, 7, 8a, 8b, 10c |
| 2 | Sauvageau & Yesovitch (2006) | 58 | Man | NR | Schizophrenia | NR | No hoarding behaviour | Suicidal asphyxia by toilet paper | 12/13 | 5c, 5d |
| 3 | Saint-Martin, Bouyssy & O'Byrne (2007) | 30 | Man | NR | BPD | Psychotropic drugs: antidepressant, tranquilizers and conventional antipsychotics | No hoarding behaviour | Suicidal asphyxia by toilet paper ingurgitation | 12/13 | 5c, 5d |
| 4 | Saint-Martin, Lefrancq & Sauvageau (2012) | 91 | Woman | NR | Alzheimer's disease | NA | No hoarding behaviour | Homicidal asphyxia by toilet paper | 12/13 | 5c, 5d |
| 5 | Fisher et al. (2014) | 30 | Man | NR | Pica | NR | No hoarding behaviour | Gastritis by toilet paper ingestion (Pica) | 10/13 | 5d, 7, 10c |
| 6 | Chisholm & Martin (1981) | 37 | Woman | NR | Pica | Zinc and ferrous sulfate | No hoarding behaviour | Pica by toilet paper ingestion | 12/13 | 5c |

Notes:
[*] Checklist items from CARE guidelines include: 1, 2, 3a, 3b, 3c, 3d, 4, 5a, 5b, 5c, 5d, 6, 7, 8a, 8b, 8c, 8d, 9a, 9b, 9c, 10a, 10b, 10c, 10d, 11a, 11b, 11c, 11d, 12, 13.
[#] Items that are not applicable for the case report are not included in this section.
Abbreviations: BDP, borderline personality disorder; OCD, obsessive compulsive disorder; NA, not applicable; NR, not reported; tACS, transcranial alternating current stimulation; y.o., years old.

Most clinical studies suggest that up to 10% of patients suffering from COVID-19 suffer from diarrhoea (Chen et al., 2020c; Guan et al., 2020; Huang et al., 2020; Jin et al., 2020; Kim et al., 2020; Li et al., 2020b; Liu et al., 2020; Xu et al., 2020b; Zhang et al., 2020b), although some studies reported higher rates, between 15% and 34% (Chen et al., 2020b; Pan et al., 2020; Wang et al., 2020a; Zhao et al., 2020). One study (Lei et al., 2020) comparing the clinical features of patients with COVID-19 in Wuhan and outside Wuhan (Guangzhou, China) reported a greater proportion of diarrhoea in the subsample of patients outside Wuhan (25% vs. 2%). Another study suggests that the prevalence of diarrhoea is greater (18.8%) in hospitalised frontline medical workers from Wuhan (Wang et al., 2020b). A recent meta-analysis that included 58 studies with COVID-19 patients

with data on the prevalence of diarrhoea reported a pooled prevalence of diarrhoea of 12.5% (95% CI [9.6–16.0]) (*Cheung et al., 2020*). A similar prevalence (12.9%) was also reported by another meta-analysis including 24 studies (*Zhu et al., 2020*). Other studies in European countries have found even higher rates of diarrhoea, up to half of patients (*Klopfenstein et al., 2020*; *Lechien et al., 2020*).

A study exploring the clinical characteristics of COVID-19 patients without or with gastrointestinal symptoms (nausea, vomiting or diarrhoea) suggests that the gastrointestinal expression of symptoms is associated with some risk factors (family clustering in exposure, pre-existing chronic liver disease) and with a more severe/critical type of the disease and higher rates of body temperature >38.5 °C (*Jin et al., 2020*). However, the association between diarrhoea and greater disease severity has not been a well-replicated finding, and meta-analysis suggests that there is no relationship between this gastrointestinal symptom and the severity of the COVID-19 disease (*Henry et al., 2020*). Another study pointed out that 19.4% of COVID-19 patients with gastrointestinal symptoms experienced diarrhoea as their first symptom before the onset of respiratory symptoms (*Han et al., 2020*).

The SARS-CoV-2 protein interacts with human angiotensin-converting enzyme 2 (ACE2) molecules, which are highly expressed in absorptive enterocytes from the ileum and colon (*Adhikari et al., 2020*; *Zhang et al., 2020a*). ACE2 is recognised as an important regulator of intestinal inflammation, and it has been hypothesised that this is the causal mechanism of diarrhoea in COVID-19 (*Ong, Young & Ong, 2020*). The SARS-CoV-2 binding affinity for human ACE2 is significantly stronger (10–20 times greater) than its 2003 SARS-CoV predecessor (*D'Amico et al., 2020*). Already in February, some authors suggested that faecal-oral transmission of SARS-CoV-2 was possible (*Yeo, Kaushal & Yeo, 2020*), with later studies confirming the presence of SARS-CoV-2 RNA in stool specimens of approximately 53–66% of patients (*Chen et al., 2020a*; *Xiao et al., 2020*), independent of the presence of gastrointestinal symptoms or the severity of illness (*Chen et al., 2020a*). There have been cases in which the SARS-CoV-2 test was negative in the nasopharyngeal swab test after treatment but the rectal test swab specimens still tested positive (*Wei et al., 2020*), particularly in paediatric patients (*Xu et al., 2020a*), suggesting that the rectal swab may be equally as important as the pharyngeal swab (*He et al., 2020*). Surveillance and adequate disinfection in latrines in areas with severe SARS-CoV-2 infection to avoid fomite transmission have also been recommended by some authors (*He et al., 2020*), as well as avoiding sharing toilets with families for patients with COVID-19 when discharged to home (*Li et al., 2020a*). As upper gastrointestinal endoscopy can induce coughing and lower gastrointestinal endoscopy can generate aerosol droplets as air is expelled from patients, preparedness for personal protective equipment in the endoscopy setting has also been recommended (*Ong, Young & Ong, 2020*; *Wong, Lui & Sung, 2020*).

Previous research has not detected viral RNA in urine specimens (*Wang et al., 2020c*). We did not find studies reporting a direct effect of SARS-CoV-2 on polyuria. However, it is important to underscore that in patients with diabetes, COVID-19 might induce diabetic ketoacidosis (*Li et al., 2020c*), which is a cause of polyuria.

Finally, no studies about toilet paper usage or hoarding in patients with COVID-19 were found.

Social cognitive biases and social media as facilitators of toilet paper hoarding (Mechanism #2)

Social cognitive biases might contribute to the mimicking of conduct by other people. A particularly pivotal role in socially replicated conduct is the bandwagon effect, which might be defined as a phenomenon where the rate of uptake of beliefs, ideas, fads and trends increases the more that they have already been adopted by others (*O'Connor & Clark, 2019*). This effect has been applied in politics since the 19th century, with the term 'jump on the bandwagon' coined when the circus clown Dan Rice used a bandwagon for the political campaign of future-president Zachary Taylor (*Chappelow, 2019*). This effect might be used to explain some behaviours, such as buying paper toilet rolls if everyone else is buying them. In fact, toilet paper hoarding is a phenomenon that has been proven to be sensitive to this bandwagon effect in other time periods. For instance, in December 1973, in a time of shortages in the United States due to the OPEC oil embargo, Johnny Carson made a joke during his opening monologue of The Tonight Show about an upcoming toilet paper shortage and triggered a nationwide toilet paper buying spree (*Malcom, 1974*). Moreover, stress-related situations might also be involved (e.g. the oil crisis in 1973 and the COVID-19 pandemic in 2019–2020), as stress is thought to potentiate decision biases along with a shift from deliberative to intuitive thinking (*Yu, 2016*; *Jacob et al., 2017*). Information bias during decision making favours considering the benefits of saving and the costs of discarding, which can lead to hoarding behaviour (*Steketee & Frost, 2003*). As already mentioned, the study by *Laato et al. (2020)* suggests that exposure to online information might contribute to increased buying behaviour and hoarding during the COVID-19 pandemic. The findings of this study also suggest that the intention to self-isolate was a major reason why people made unusual purchases during COVID-19 (to prepare for isolation and quarantine).

People with acute stress disorder report more cognitive biases pertaining to external harm, somatic sensations and social events (*Smith & Bryant, 2000*), suggesting that stress moderates reasoning capability. Socially anxious people are more prone to interpret emotionally ambiguous situations as threatening or negative, also known as interpretation bias, which is involved in the maintenance of anxiety and stress reactivity (*Badra et al., 2017*; *Van Bockstaele et al., 2019*). Some authors have suggested that people with elevated negative affectivity and social inhibition, also known as type D personality (*Denollet, 2005*), might perceive greater threat and report stronger feelings of distress during ambiguous situations (*Grynberg et al., 2012*) and exhibit an increased risk of stress-related cardiovascular events (*Denollet et al., 2006*). Studies suggest that individuals with high social stress tend towards vigilance with regard to subliminal social threat cues but not subliminal physical threat cues (*Helzer, Connor-Smith & Reed, 2009*). A general negative cognitive bias when coping with traumatic exposures is considered to be a risk factor for post-traumatic stress disorder (*DiGangi et al., 2013*). Traumatic life experiences have also been suggested to increase psychosis proneness via cognitive biases (*Gawęda et al., 2018*), such as jumping to conclusions ('not needing long to reach a

conclusion'), belief inflexibility bias ('not needing to consider alternatives when making a decision'), attention to threat bias ('people cannot be trusted') and external attribution bias ('things go wrong because of other people'). Previous research exploring the response to social stress in a virtual reality environment suggests that there is an additive effect of separate cognitive biases on paranoid responses to social stress, with greater effects via attention to threat bias and external attribution bias (*Pot-Kolder et al., 2018*). Studies including patients with schizophrenia and acute delusions also indicate that patients under stress show an increased bias of jumping to conclusions (*Moritz et al., 2015*).

Risk communication, defined by the World Health Organization as 'the exchange of real-time information, advice and opinions between experts and people facing threats to their health, economic or social well-being,' might lead to hoarding behaviour (*Abrams & Greenhawt, 2020*). This risk communication has become more relevant in recent years, as social media networks are constantly increasing. Previous studies modelling the propagation of social responses during a disease outbreak (*Fast et al., 2015*), which include the hoarding of medical supplies, suggest that heightened social responses spread through the population via two mechanisms: (1) when a disease is novel to the region or is perceived as particularly threatening, media influence spreads concern through the population; (2) when communicating with their neighbours, agents are biased towards adopting the opinions of their more concerned neighbours rather than the most calm ones.

Another threat to human society is digital misinformation, which has been suggested to be related to the phenomenon called 'echo chambers,' leading to diffusion with a bandwagon effect (*Törnberg, 2018*). Another problem of misinformation is that false news diffuses faster than true news in social networks (*Vosoughi, Roy & Aral, 2018*). The bandwagon effect does not apply only to negative or threatening news. For instance, during the COVID-19 pandemic, the toilet paper challenge spread over social media and was replicated by thousands of people. This challenge, also known as the '10 Touch Challenge,' was initially proposed by football players who tried to juggle a roll of toilet paper ten times with their feet, similar to how soccer players juggle soccer balls in training (*White, 2020*). Thousands of people uploaded their personal videos on the internet, which seemed to relieve the negative effects of the lockdown because most people ended their videos with a satisfied smile. Although it is unknown how long this positive psychological effect lasts, this conduct clearly reflects how the bandwagon effect contributed to the inadequate use of toilet paper during the COVID-19 pandemic.

Stress worsens mental health and toilet paper hoarding (Mechanism #3)

Stress promotes the secretion of hormones (e.g. glucocorticoids, catecholamines) that are adaptive in the short term but that might promote pathophysiological processes over longer time periods, when they are secreted in excess or are dysregulated either by not being produced in sufficient amounts during periods of challenge or change or by not being turned off efficiently after the challenge (*McEwen, 2001*). Bruce McEwen coined the term allostatic load to define 'the wear and tear on the body' as a result of the accumulation of chronic stress (*McEwen, 1998*). This model might be applied to most

mental illnesses, including mood disorders (*McEwen, 2003*), psychotic disorders (*Nugent et al., 2015*) and anxiety disorders (*Nolte et al., 2011*).

Stressful and traumatic life events might trigger the onset of hoarding disorder, particularly for cases with a later onset (*Tolin et al., 2010*; *Landau et al., 2011*). Stress, mainly changes in relationships and interpersonal violence, are also associated with an exacerbation of hoarding behaviour (*Tolin et al., 2010*). Other studies point out that early life stress with insecure attachment (*Danet & Secouet, 2018*; *Crone et al., 2019*) or low parental emotional warmth (*Alonso et al., 2004*) might play a role in the pathogenesis of hoarding behaviours. Traumatic life events are associated with a greater severity of hoarding symptoms, particularly in the clutter factor of compulsive hoarding (but not in the difficulty of discarding or acquisitioning) (*Cromer, Schmidt & Murphy, 2007*). It has been suggested that the coexistence of traumatic experiences and inattention and hyperactivity symptoms could contribute to the difficulties of clutter and organisation reported by hoarders (*Hartl et al., 2005*). However, other experimental studies that have tested whether stress influences saving and acquiring behavioural tendencies in young adults (*Shaw & Timpano, 2016*) have yielded unexpected results: participants in the stress condition saved and acquired fewer items than those in the control condition. As discussed by the authors of the previous study (*Shaw & Timpano, 2016*), the laboratory stressor may not have been strong enough to increase saving and acquiring behavioural tendencies, and there is a need to conduct studies exploring the effects of acute stressors that are more similar to real-life stressors experienced by individuals with hoarding (such as interpersonal conflict).

Intolerance to uncertainty has been proposed as a risk factor for hoarding behaviour (*Wheaton et al., 2016*). Interestingly, recent studies exploring the role of intolerance to uncertainty in mental well-being associated with the COVID-19 pandemic have reported that the combination of rumination and fear of COVID-19 mediates the association between intolerance to uncertainty and mental well-being (*Satici et al., 2020*). Intolerance to uncertainty is also a predictor of the severity of hoarding symptoms in people with hoarding disorder (*Worden et al., 2019*). Many of the recommended measures during the COVID-19 pandemic, such as washing and prevention of contamination as well as the quarantine and nationwide lockdown, are thought to worsen symptoms of patients with OCD or hoarding behaviours (*Banerjee, 2020*). Recent preliminary studies suggest that OCD patients experienced worsened symptoms, particularly contamination obsessions, during the COVID-19 pandemic (*Davide et al., 2020*).

Personality traits are also important moderators of the response to stressful situations, particularly neuroticism, which appears to play a prominent role in the stress process (*De Jong, Van Sonderen & Emmelkamp, 1999*). People with high neuroticism report more exposure to stressors (*Bolger & Schilling, 1991*), higher perceived stress (*Ebstrup et al., 2011*; *Kim et al., 2016*) and more inadequate coping strategies (*Connor-Smith & Flachsbart, 2007*). People with high neuroticism are also at greater risk for major depression and more sensitive to the depressogenic effects of adversity resulting from exposure to stressful life events (*Kendler, Kuhn & Prescott, 2004*). Neuroticism has also been associated with hoarding obsessions and compulsions in a study that assessed personality with the NEO

Personality Inventory–Revised (*LaSalle-Ricci et al., 2006*). In this later study, hoarding was negatively correlated with conscientiousness.

Regarding the COVID-19 pandemic, there are four studies that have analysed the role of personality traits in toilet paper stockpiling. The first study by *Garbe, Rau & Toppe (2020)*, already mentioned in the Results section of the systematic review, reported that conscientiousness was associated with toilet paper stockpiling, although emotionality had an indirect effect on stockpiling by means of the threat of COVID-19. Another unpublished study by *Columbus (2020)* conducted a survey in two samples of United Kingdom (UK) residents and considered the stockpiling of foods or supplies. Approximately 36% (sample 1) to 40% (sample 2) of participants reported having bought more food or supplies than they usually did during the preceding two weeks in response to the COVID-19 pandemic. Honesty–humility showed a negative association with past stockpiling (sample 1) and a positive association with intentions to refrain from stockpiling in the future (sample 2). The association between this personality dimension and stockpiling was not mediated by beliefs about the shopping behaviour of others. However, other studies suggest that viewing others experiencing stress creates a 'contagious' physiological stress response, with faster responses in people with high dispositional levels of empathy (*Dimitroff et al., 2017*). A recent study (*Zettler et al., 2020*) explored the relationship between nine personality factors (including HEXACO and Big Five personality traits) and hoarding behaviour during the COVID-19 pandemic in five independent samples from two Western European countries (overall sample: $N = 10.702$). In this study, honesty-humility and agreeableness personality traits were negatively associated with hoarding behaviour.

Another study (*Bentall et al., 2020*) explored over-purchasing using data collected in the early stages of the COVID-19 pandemic from two large population internet surveys in the UK and the Republic of Ireland. People did not over-purchase toilet paper more than other common supplies (e.g. tinned food or dried foods). However, this study did not specifically explore psychological determinants of toilet paper hoarding or toilet paper over-purchasing when compared to other items. Over-purchasing or hoarding was found to be positively associated with household income, the presence of children at home, depression, anxiety and mistrust of others or paranoia. Regarding personality traits, conscientiousness was negatively associated with over-purchasing in both samples. In the Irish sample, openness was also negatively associated with over-purchasing in the Irish sample. In the UK sample, extraversion was associated with over-purchasing, whereas neuroticism was negatively associated with over-purchasing. The variables were found to predict approximately 34–36% of the variance of the model in the Republic of Ireland and the UK.

One study (*Bai, 2020*) using two datasets from the UK (cross-sectional) and the United States (longitudinal) tested whether people who endorse conspiracy theories may be particularly likely to engage in panic buying behaviours during the COVID-19 pandemic. The study found a positive relationship between conspiracy theory endorsement and stockpiling behaviour in both samples. In the US sample, believing that COVID-19 is a real threat was another predictor of stockpiling. Longitudinal analyses

suggested that conspiracy theory endorsement was a predictor of stockpiling behaviour in the future, even after controlling for self-reported baseline stockpiling behaviours.

Psychoanalytical explanations for the hoarding of toilet paper might be formulated, such as a form of regression to the anal stage allowing our ego to feel in control of an incontrollable situation (COVID-19 pandemic) (*Anghelou, 2020*; *Wood, 2020*). As suggested by Freud, the second stage of psychosexual development is the anal stage (typically occurring during the 2nd year of life), in which the child's interest and sexual pleasure are focused on the expulsion and retention of faeces and the sadistic instinct is linked to the desire to both possess and destroy the object (*American Psychological Association, 2020*). Some authors (*Güzel, 2020*) think that panic and restlessness over toilet paper was a response to political failure and that toilet panic hoarding might be understood as a crude solution of the overwhelmed and fragmented subject in the absence of a symbol of authority. Psychoanalytic theories also suggest that a regression to the anal phase might occur in people with hoarding disorder, particularly when a traumatic or emotionally distressing event happens (*Camps & Le, 2019*).

Cultural aspects moderate the relationship between the COVID-19 pandemic and toilet paper hoarding (Mechanism #4)

Some studies have explored whether hoarding disorder features differ across distinct cultural settings. A study that included patients with hoarding disorder from the United Kingdom, Spain, Japan and Brazil (*Nordsletten et al., 2018*) indicates that the severity and core features of hoarding disorder as well as the cognitions and behaviours commonly associated with this condition are largely stable across cultures. One study comparing symptoms from the hoarding dimension in patients with OCD from China, the USA and Brazil reported a lower proportion of hoarding symptoms in the sample of patients from China (*Li et al., 2009*). However, another study found that hoarding disorder in East Asia is relatively common and symptomatically similar to that reported in Western countries (*Wang et al., 2016*).

Other studies have explored potential cultural differences in cognitive biases. In a study that examined the relationship between interpretation bias and social anxiety among Chinese adolescents, the results were similar to those found in Western samples (*Yu et al., 2019*). Although studies have not addressed whether there are differences in the social response to the COVID-19 pandemic by distinct countries or cultures, an indirect way to approach this question is to explore Google trend topics. In the Google trend topics by country for the word 'toilet paper,' Australia was the leading country (score of 100), followed by the USA (score of 74) and Canada (score of 42). The trend in the use of the search term 'toilet paper' on Google was similar for these three countries and the United Kingdom (Figs. 2A–2D), although a different pattern was observed for India (Fig. 2E), another country in which English is an official language. As shown in Fig. 2, most countries had a peak in March 2020, which coincides with the COVID-19 outbreaks in different countries and the implementation of lockdowns. The massive search for 'toilet paper' decreased in a few weeks to previous levels. If we compare the Google search trends for 'toilet paper' and 'COVID,' in most countries, the 'COVID' term was always ranked above 'toilet paper' in search interest (Fig. S1), with the exception of Australia, such

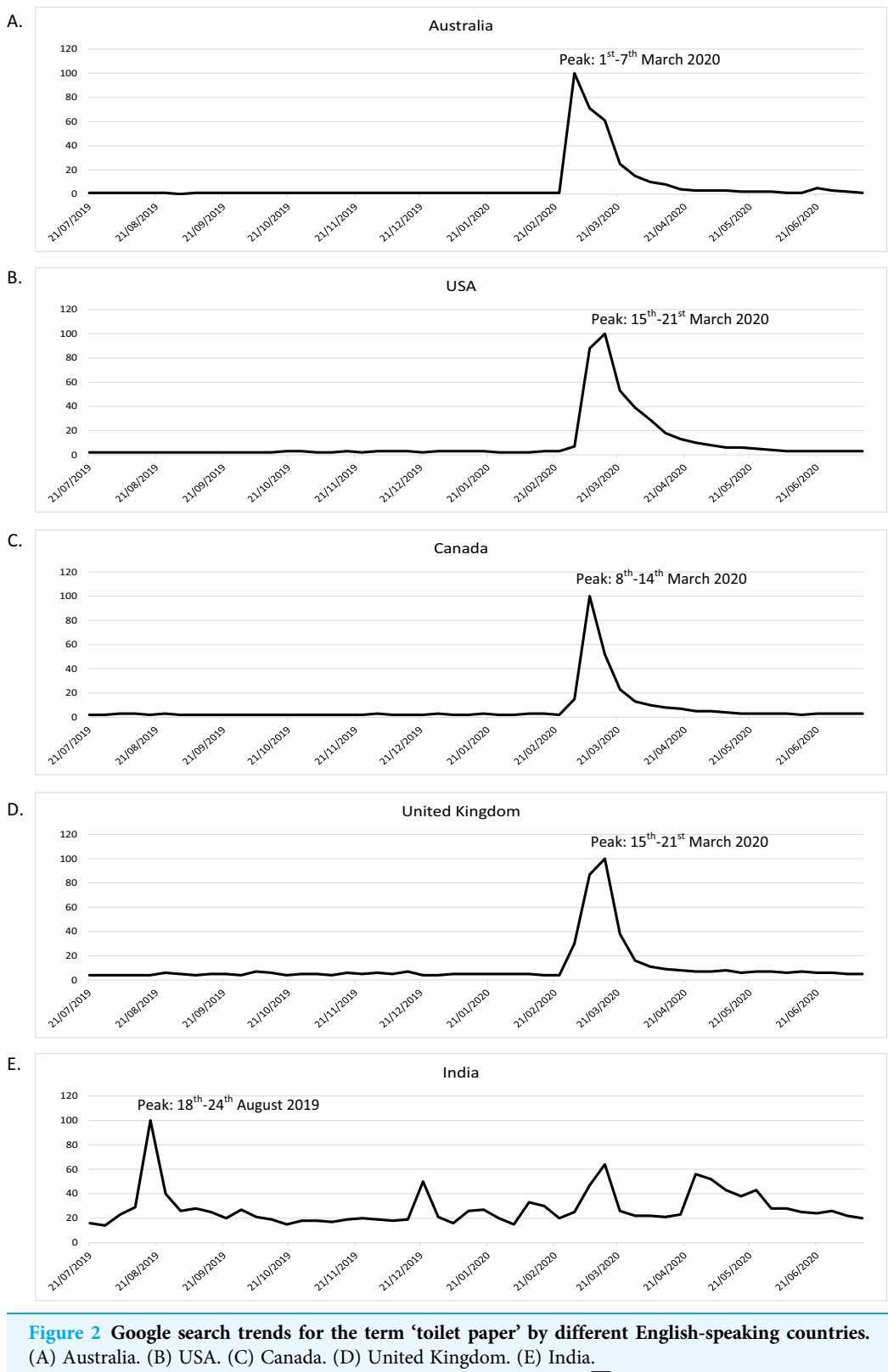

**Figure 2 Google search trends for the term 'toilet paper' by different English-speaking countries.**
(A) Australia. (B) USA. (C) Canada. (D) United Kingdom. (E) India.

that in the first week of March, the interest in 'toilet paper' was 20, clearly above interest in the term 'COVID' (5).

In a previous study that explored panic buying of toilet paper (*Keane & Neal, 2020*), an index of panic during the COVID-19 pandemic was created considering five terms: toilet paper, panic buying, hoarding, panic, and supermarket. For non-English-speaking countries, these terms were translated. Countries were grouped into three regions (Europe and North America; Asia [including Oceania]; and the rest of the world). *Keane & Neal (2020)* found significant heterogeneity between regions in the timing and severity of panic between January and April 2020. They also compared the peak panic indexes between countries: Italy (panic index of 0.15 on 22 March 2020, following the national lockdown on 20 March 2020); France (panic index of 0.083 on 16 March 2020, the same day of the announcement of their nationwide lockdown); United Kingdom (panic index of 0.18 on 22 March 2020, occurring in the same week of the announcement of internal restrictions, including school closings and restrictions on gatherings and movement); and Australia (panic index of 0.79 on 4 March 2020; it was the country with the greatest speed of panic spread (as the panic index was 0.08 2 days before, on 2 March 2020)). As there were no important policy announcements in Australia by this time (restrictions on gatherings were announced on 13 March 2020), it is difficult to explain this massive spike with these factors. The authors of the last study concluded that their model could not explain this panic pattern in Australia. Other countries with massive spikes that could not be easily explained were Japan, Taiwan and Singapore. The study of Keane and Neal suggested that internal movement restrictions generate considerable consumer panic in the short term, but the effect largely vanishes after a week to ten days. Moreover, they also found a response of consumer panic to announcements of internal movement restrictions in foreign countries.

A cultural aspect that is important to consider is the use of toilet paper by different countries. Despite the global reduction in open defecation in the last two decades, which might be defined as the lack of use of toilet facilities for defecation, there is still a substantial worldwide proportion of people from rural areas in less economically developed countries who engage in open defecation. For instance, World Bank data suggest that the prevalence of open defecation in rural areas worldwide was 37.1% in 2000, with a reduction of up to 18.3% in 2017 (*World Bank Group, 2000a*). This information points to possible differences in the use of toilet paper and likely hoarding behaviour aiming at conserving this item in rural areas when compared to urban areas, as only 1.5% of the population of urban areas continues to engage in open defecation (*World Bank Group, 2000b*).

Toilet paper consumption differs by country. The estimated annual per capita toilet paper consumption in selected countries in 2018 (obtained from Statista Consumer Market Outlook) (*Armstrong, 2020*) describes the USA as the leading country (141 rolls and 12.7 kg), followed by Germany (134 rolls and 12.1 kg) and the UK (127 rolls and 11.4 kg). There might also be differences in cleaning habits between people from different countries. For instance, data from a WIN/Gallup International survey conducted in 2015 suggest that only 50% of people in the Netherlands wash their hands with soap and

water after using the toilet, compared to 96% of people in Bosnia and Herzegovina (*Marian, 2015*).

## DISCUSSION

Our study aimed to explore the potential contribution of the COVID-19 pandemic to toilet paper hoarding. Our systematic review highlights the scarcity of studies addressing this important topic, and we identified very little published data. We want to highlight the study by *Garbe, Rau & Toppe (2020)* as being unique because they added empirical data on the influence of the perceived threat of COVID-19 and personality traits (mainly conscientiousness) on several behavioural aspects related to toilet paper shopping and stockpiling. Other studies were focused on the hoarding of supplies (*Columbus, 2020*; *Oosterhoff & Palmer, 2020*), and they were not specifically focused on toilet paper hoarding, such as the study by *Garbe, Rau & Toppe (2020)*. The secondary outcome of our systematic review focused on mental health, and the pathological use of toilet paper also underscores that this is an under-researched topic, as we could identify only six case reports regarding OCD, suicide, homicide or pica, although the quality of the case reports was relatively good. The methodology of a realist review allowed the study of potential mechanisms contributing to toilet paper hoarding in the COVID-19 pandemic.

### Potential mechanisms relating the COVID-19 pandemic to toilet paper hoarding

Although the authors of a systematic review on gastrointestinal symptoms in COVID-19 (*Miri et al., 2020*) suggested that the coexistence of diarrhoea could explain the coronavirus panic buying of toilet rolls, this hypothesis has not been adequately tested in the literature. Moreover, the presence of diarrhoea or the prolonged dissemination of SARS-CoV-2 in the faeces were lesser-known characteristics of the disease at the beginning of the outbreak, when people were buying and hoarding toilet paper. Indeed, the knowledge that there might be faecal-oral transmission of SARS-CoV-2 might induce some people to increase the use of toilet paper, but it does not seem to be the main mechanism explaining the global shopping frenzy at supermarkets. The relatively low proportion of diarrhoea (approximately 12–13%) found in people with COVID-19 infection does not seem to justify the global trends in shopping for toilet paper. Moreover, shopping for and hoarding of toilet paper appeared to be more intense in the first weeks following the COVID-19 outbreak all around the world, with a reduction in the following weeks. This generalised behaviour in stores seems to mimic the Google trend surge on the internet for the word 'toilet paper' during March 2020 and was amplified by the national lockdowns in most but not all (e.g. Australia) countries (*Keane & Neal, 2020*).

The mechanism linking social cognitive biases seems to contribute to hoarding behaviour more clearly than the gastrointestinal mechanism. The bandwagon effect is likely the most replicated bias in different countries, as this effect has been previously found to be associated with toilet paper buying (*Malcom, 1974*). The progressive increase in social networks also seems to have contributed to the fast and worldwide expansion of toilet paper hoarding due to this cognitive bias, with this behaviour being replicated in

many countries. Other negative affect and interpretation biases might be linked to intolerance to uncertainty, a clinical characteristic that has been associated with hoarding behaviour (*Wheaton et al., 2016*). These biases might be even more important given the uncertainty of the COVID-19 situation (*Koffman et al., 2020*), as the SARS-CoV-2 virus is a new virus with much information to be discovered. Interestingly, the intolerance of uncertainty was associated with poorer mental well-being mediated by both the fear of COVID-19 and rumination (*Satici et al., 2020*).

Another question to be resolved is whether risk factors for toilet paper hoarding during the COVID-19 pandemic are shared with other hoarding behaviours. In this sense, one study pointed out that conscientiousness is a personality trait linked to toilet paper stockpiling during the COVID-19 pandemic (*Garbe, Rau & Toppe, 2020*), whereas other studies including clinical samples of patients with hoarding symptoms found an opposite result (lower conscientiousness associated with hoarding symptoms) (*LaSalle-Ricci et al., 2006*). The different roles of conscientiousness in patients with hoarding symptoms and healthy people who hoarded toilet paper during the COVID-19 pandemic is an interesting finding that merits some discussion. Conscientiousness is a personality trait that implies being more efficient and organised, showing self-discipline that involves planned behaviour (*Costa, McCrae & Dye, 1991*). This personality trait fits well with the idea that healthy people under a stressful situation (e.g. COVID-19 pandemic) might decide to buy and hoard toilet paper, particularly when news points to the possibility of a shortage of toilet paper (*Schrotenboer, 2020*). Although some studies have related conscientiousness with OCD (*Rector et al., 2002*; *Inchausti, Delgado & Prieto, 2015*), other studies have found lower conscientiousness in OCD patients than in healthy controls (*Hwang et al., 2012*). Moreover, other studies suggest that there might exist differences based on the OCD phenotype: higher conscientiousness in comorbid tic-related OCD (*Nestadt et al., 2009*) and lower conscientiousness in a comorbid affective-related class (*Nestadt et al., 2009*) or with the presence of hoarding symptoms (*LaSalle-Ricci et al., 2006*; *Samuels et al., 2008*; *Boerema et al., 2019*). The different associations between conscientiousness and hoarding behaviour in non-clinical (higher conscientiousness) and clinical populations (low conscientiousness) is an intriguing finding, as the non-clinical study included people recruited during the COVID-19 pandemic (*Garbe, Rau & Toppe, 2020*), whereas the clinical studies included patients with OCD (*LaSalle-Ricci et al., 2006*; *Samuels et al., 2008*; *Boerema et al., 2019*). Two studies found a negative association between honesty-humility and hoarding food and supplies (*Columbus, 2020*; *Zettler et al., 2020*). Although no previous studies have explored honesty-humility personality traits in clinical samples of patients with hoarding disorder, this personality trait is associated with trustworthiness (*Thielmann & Hilbig, 2015*) and cooperation with others. People with hoarding symptoms show increased feelings of hostility in response to social exclusion (*Mathes et al., 2019*); therefore, it could be hypothesised that hoarders might have a reduced tendency to cooperate with others. It is notable that the two studies reporting associations between honesty-humility and hoarding behaviour during the COVID-19 pandemic (*Columbus, 2020*; *Zettler et al., 2020*) did not differentiate the subtype of stockpiled items when exploring the contribution of personality factors.

In the study by *Garbe, Rau & Toppe (2020)*, which was focused on toilet paper, honesty-humility was not associated with toilet paper stockpiling.

It is possible that hoarding of toilet paper is a distinct phenotype compared with hoarding other items, at least in terms of neurobiological/psychological pathophysiological pathways. This important question has yet to be answered, as studies focused on toilet paper hoarding are scarce. Future studies might examine whether personality traits linked to hoarding differ based on the subtype of hoarded items. Although speculative, it could be that toilet paper hoarding is a distinct subtype of hoarding disorder. To date, no definitive conclusions can be drawn, and more research needs to address this issue before assuming a different subtype of hoarding disorder or even considering the inclusion of a specifier for toilet paper hoarding in future diagnostic classifications (e.g. DSM-6). Another limitation of previous research on toilet paper hoarding during the COVID-19 pandemic is that most of the data come from surveys without the administration of diagnostic interviews by a psychiatrist or a clinical psychologist. Therefore, it is important to conduct clinical studies in the future to scrutinize the potential boundaries between mental illnesses and non-psychiatric conditions in the research of toilet paper hoarding. Although the diagnosis of a mental illness might require dysfunction criteria, the study of the boundaries of psychiatric illnesses may not be resolved until there is a detailed understanding of the pathophysiology of the disorders (*Kendell & Jablensky, 2003*).

Future studies also need to better address potential cultural differences that could explain some differences in toilet paper hoarding between countries. An intriguing question is why Australians were the leaders in panic buying. Tim Neal, who participated in a study about panic buying during the COVID-19 pandemic (*Keane & Neal, 2020*), pointed out that the Australian media's coverage of hoarding could have contributed to the world-leading levels of panic (*Zhou, 2020*). Other Asian countries, such as Japan, Taiwan and Singapore, that also had massive spikes that could not be easily explained were found in the model developed by *Keane & Neal (2020)*. Shocking news from Asian countries was also reported early in the COVID pandemic, including an armed robbery of toilet paper in Hong Kong (*Ho-Him, 2020*) or the chaining of toilet paper rolls in public toilets in Japan (*Acharya, 2020*). Some authors have suggested that the dense, close-knit networks of some countries (e.g. Singapore) make people more prone to adopt the fears and behaviours of the people around them (*Bouffanais, 2020*).

## Managing toilet paper hoarding: a proposed algorithm from the CATOTIM group

The management of potential cases of toilet paper hoarding is a challenge for the clinician. The differential diagnosis of a patient with hoarding symptoms is quite complex because hoarding symptoms might be present in different psychiatric and neurological conditions (*Pertusa et al., 2010*) and because patients with hoarding disorder often underreport specific symptoms (*DiMauro et al., 2013*). Recent epidemiological studies indicate that the prevalence of hoarding disorder in the general population is 2.5% (confidence interval: 1.7–3.6%), with similar prevalence rates for both males and females (*Postlethwaite, Kellett & Mataix-Cols, 2019*). We have tried to integrate the main findings

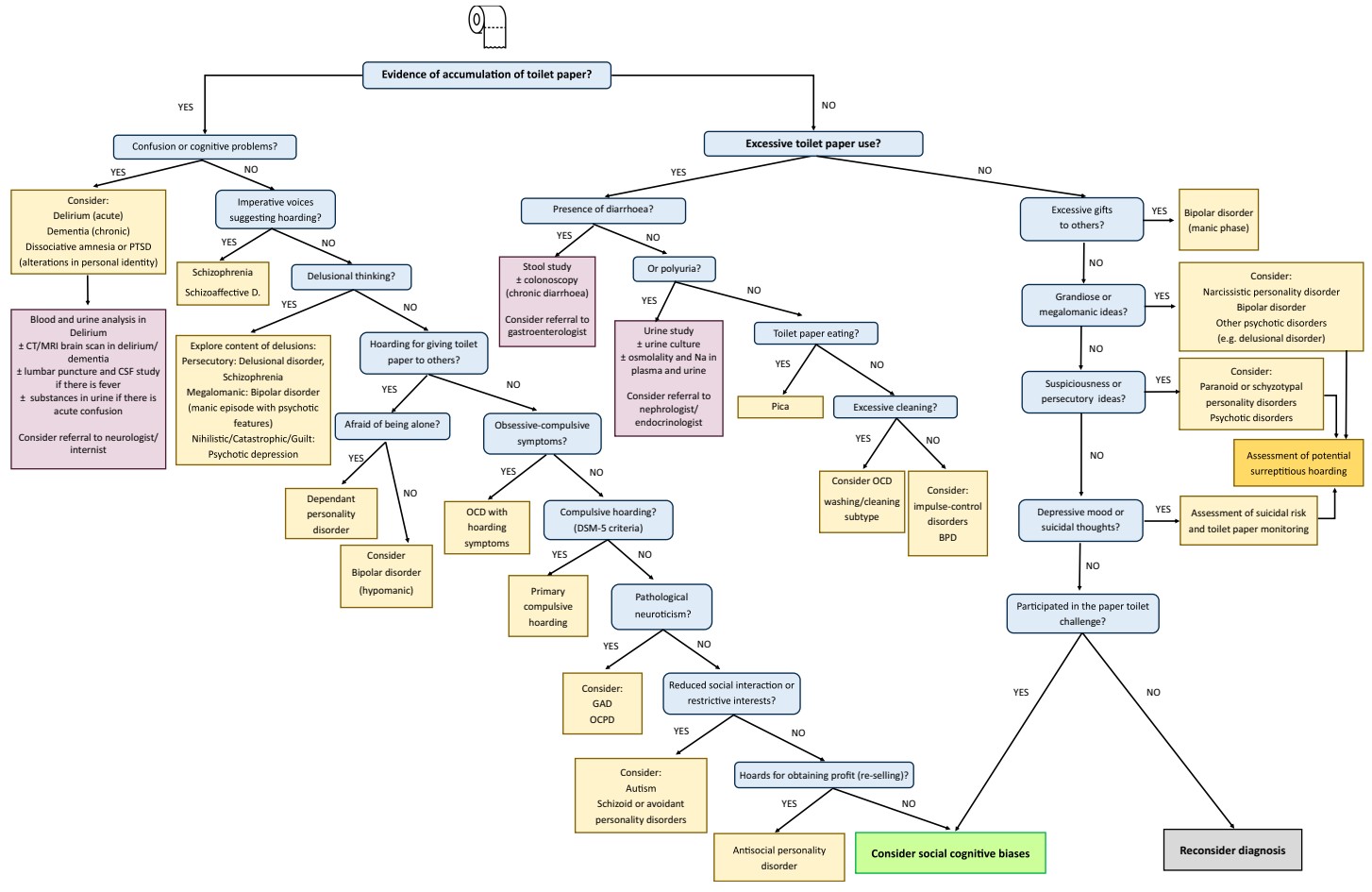

**Figure 3 CATOTIM algorithm for managing paper toilet hoarding.** Abbreviations: CATOTIM, catalan toilet tissue research group in mental health; PTSD, post-traumatic stress disorder; CT, computed tomography; MRI, magnetic resonance imaging; CSF, cerebrospinal fluid; Shizoaffective D. Schizoaffective disorder; OCD, obsessive-compulsive disorder; DSM-5, diagnostic and statistical manual of mental disorders—fifth edition; GAD, generalised anxiety disorder; OCPD, obsessive-compulsive personality disorder.

of our review and the personal expertise of the members of the Catalan Toilet Tissue Research Group in Mental Health (CATOTIM) that participated in this study in a proposed algorithm that is described in Fig. 3. A validation study for this algorithm has not been included; therefore, our pilot algorithm for managing toilet paper hoarding should be considered a theoretical proposal. The generation of the algorithm was a dynamic process. Successive versions of the algorithm were created taking into account the findings of the systematic and realist reviews and the comments from all CATOTIM members.

As shown in Fig. 3, the first key question is to know whether there is an accumulation (hoarding) of toilet paper. For those cases with evident toilet paper hoarding, psychopathological assessment needs to first detect potential confusion or cognitive problems (attention deficits, memory loss). In that case, it is important to eliminate the possibility of neurological syndromes such as dementia that have been reported to be associated with hoarding symptoms in approximately 23–29% of cases (*Hwang et al., 1998*; *Mitchell et al., 2019*). Patients suffering from delirium might have complex stereotyped

movements and, rarely, the mimicking of a work pattern (occupational delirium) (*Burns, Gallagley & Byrne, 2004*). In these situations, it is important to disregard intercurrent medical processes, and it might be necessary to perform blood and urine tests, CT or MRI brain scans, substance use studies, and/or cerebrospinal fluid analyses (in cases with fever). In those patients accumulating toilet paper who show amnesia of the situation and alterations in personal identity, dissociative disorders including post-traumatic stress disorder need to be considered. For this reason, inquiry about potential toilet-related traumatic events may shed light on this diagnosis. In oriented patients, the presence of specific symptoms might lead to specific diagnoses: auditory hallucinations in patients with schizophrenia or schizoaffective disorders and specific delusions in patients with non-affective (e.g. schizophrenia) or affective (bipolar disorder, psychotic depression) psychoses. For instance, a patient suffering from major depression with psychotic features might hoard paper if there are nihilistic or catastrophic delusional ideas (e.g. the belief that bad things are about to happen, feelings of being rotten) (*Rothschild, 2013*).

In some cases, it is possible that people hoard paper to give it to others. In cases when there is a long-standing need for the person to be taken care of and a fear of being abandoned or separated from close individuals, the possibility of a dependent personality disorder needs to be considered. People with bipolar disorder with hypomanic symptoms might also hoard paper for making gifts to others, although the presence of a euphoric mood could also guide the diagnosis.

A particularly important condition to be considered is OCD. Initially, hoarding symptoms were thought to be a feature of OCD, but in the last DSM-5, a distinct entity for compulsive hoarding was included. It is critical to explore other obsessive-compulsive symptoms (cleaning obsessions and washing compulsions, sexual/religious obsessions, aggressive obsessions with checking compulsions, symmetry obsessions with ordering compulsions) because their existence can guide the diagnosis to OCD when compared with a primary hoarding disorder without obsessive-compulsive symptoms (*Pertusa et al., 2008*). Notably, hoarding symptoms might be present in people with high neuroticism, particularly if they suffer from generalised anxiety disorder (*Tolin et al., 2011*) or obsessive-compulsive personality disorder (OCPD) (*Mataix-Cols et al., 2010*). If hoarding symptoms appear in people with social isolation and restricted interests, autism disorders and specific personality disorders (schizoid (indifference to social relationships, with a limited range of emotional expression and experience) and avoidant (feelings of extreme social inhibition, inadequacy, and sensitivity to negative criticism and rejection)) also need to be considered. In autism, hoarding symptoms are common (approximately 25% of cases) and are associated with internalizing and anxiety/depressive symptoms, externalizing behaviour and attention problems (*Storch et al., 2016*).

In cases in which people have been hoarding for reselling toilet paper, antisocial personality traits could be driving the hoarding. There have been documented cases of people hoarding up to 4,800 toilet paper rolls to resell them on eBay at a greater cost (*Brook, 2020*). This conduct during a pandemic shows some of the characteristics of an antisocial personality disorder (*Black, 2015*): disregard for right or wrong, deceit for exploitation of others, disrespect of others and lack of empathy for others.

If pathological conditions are not clearly found, as already mentioned, it is important to consider the contribution of social cognitive biases (e.g. bandwagon effect) for inducing hoarding symptoms in non-clinical populations.

Sometimes there is excessive buying of toilet paper secondary to excessive use without hoarding behaviour. Clinicians need to consider in these cases the potential medical causes for either diarrhoea or polyuria, with specific tests depending upon the reported symptoms. As already mentioned, in rare conditions, people might eat toilet paper secondary to pica (*Chisholm & Martin, 1981*; *Fisher et al., 2014*). In cases with pica, if hypozincaemia is observed, supplementation with zinc might resolve the abnormal eating behaviour (*Chisholm & Martin, 1981*). An excessive (pathological) use of toilet paper by OCD patients with contamination obsession symptoms needs to be considered. In some cases with resistant OCD and excessive wiping, tACS might be useful (*Klimke et al., 2016*). People with impulse-control disorders and borderline personality disorder might also use toilet paper in excess due to a lack of inhibition control.

Although in some cases there is no apparent hoarding or excessive use of toilet paper, it is important to consider the pathological use of toilet paper due to psychopathological disturbances. In people with bipolar disorder with a manic episode, spending sprees and bizarre gifts might occur. In other cases, there could be surreptitious hoarding that was not easily observed in the first assessment. This could arise in people with psychotic symptoms, particularly if there is suspiciousness (e.g. paranoid personality disorder, psychotic disorders). In a patient with depressive mood, suicidal ideation needs to be explored because toilet paper might be used as a lethal mechanism for committing suicide (*Sauvageau & Yesovitch, 2006*; *Saint-Martin, Bouyssy & O'Byrne, 2007*).

If there is no apparent psychopathology, the diagnosis might be reconsidered. However, a previous step is to be sure that the individual has not participated in the toilet paper challenge. If this is the case, it is probable that his/her conduct is driven by social cognitive biases (e.g. bandwagon effect).

Finally, it is also important to mention that under unusual circumstances, toilet paper hoarding might be considered rational behaviour. In line with this, *Laato et al. (2020)* suggest that rational decision-making processes might be affected in ill-defined, ambiguous, and unclear circumstances, such as the period in which the COVID-19 virus spread rapidly in Europe during the first wave (March 2020). As this study points out, the buying and stockpiling of toilet paper might be considered a normal behaviour in some circumstances, such as preparing for self-isolation. In these circumstances, defining pathological hoarding, when compared to excessive stockpiling, can be a difficult task.

## Clinical and ecological implications

Our study underscores the need to consider the pathological use and hoarding of toilet paper in clinical practice, as this behaviour might have negative consequences for the functioning and quality of life of people with or without serious mental illnesses. It is particularly important to eliminate the possibility of psychiatric disorders that might be associated with toilet paper hoarding and that might require specific treatments.

This approach is a challenge for psychiatrists and clinical psychologists who need to consider potential comorbid medical conditions that could also worsen this behaviour.

The potential contribution of social media to social cognitive biases (e.g. bandwagon effect) and social-driven panic behaviours underscores the importance of managing news in the media and avoiding disseminating fake news on the internet. To fight this issue, in April 2018, the European Commission and representatives of online platforms, leading social networks, advertisers and the advertising industry agreed on a self-regulatory Code of Practice to address the spread of online disinformation and fake news (*European Commission, 2020*). Attached to the principles of this Code of Practice is a step for most people using and working with social media in order to avoid the negative psychological consequences of disseminating fake news.

Recent updated analysis from the Natural Resources Defense Council (NRDC) (*Natural Resources Defense Council, 2020*) has reported the climate impacts caused by the "tree to toilet" pipeline destroying the climate-critical Canadian boreal forest. Industry is thought to clear one million acres of boreal forest each year (led by Brazil, Russia and Canada in terms of global intact forest loss) in part to produce pulp that US tissue makers roll into toilet paper (*Natural Resources Defense Council, 2020*). Environmentalists denounce this, as turning a tree into paper requires more water than turning paper back into fibre, and many brands using tree pulp also use polluting chlorine-based bleach to obtain greater whiteness (*Kaufman, 2009*). Another problem for the sustainability of the planet is the continuously growing tendency to use toilet paper (*Crumbie, 2019*). The worldwide revenue for the toilet paper segment from the tissue and hygiene paper sector in 2019 was US$ 83 billion, and it is expected to increase up to US$ 100 billion by the year 2025 (*Statista Consumer Market Outlook, 2020*). An NRDC report (*Skene, 2019*) suggests that as the market for tissue grows around the world, recycled products and alternative fibres will be the only way to accommodate increased demand without creating further strain on indigenous peoples, the climate, and biodiversity.

For all these reasons, it is important that policy makers consider the potential negative impact of toilet paper hoarding at both the individual and community levels, with potential harmful effects to the planet. Therefore, it is recommended that policy makers develop strategies that promote research on the causes and consequences of toilet paper hoarding.

## Gaps in the literature and future directions

Although a previous survey (*Garbe, Rau & Toppe, 2020*) suggested that the prevalence of toilet paper hoarding was 17.2% for North Americans and 13.7% for Europeans, more epidemiological studies are needed to weigh the real prevalence of this hoarding behaviour and to administer diagnostic interviews to eliminating the possibility of hoarding behaviour associated with psychiatric disorders or stress-related "reactive" and "transitional" conduct. Longitudinal studies could also help to explore whether these hoarding behaviours associated with the COVID-19 pandemic were only associated with the first COVID-19 outbreak or are repeated in subsequent outbreaks.

The psychological and neurobiological underpinnings of toilet paper hoarding are a fascinating field to be explored. Future research might study whether or not the mechanisms that lead to saving toilet paper are shared with other hoarded items. A particularly interesting hypothesis to be tested relies on the contribution of personality traits, given the apparent different role of conscientiousness in toilet paper hoarding during the COVID-19 pandemic (*Garbe, Rau & Toppe, 2020*) and in hoarding symptoms in people with OCD (*LaSalle-Ricci et al., 2006*; *Samuels et al., 2008*). Research on neurobiological determinants might study the contribution of stress-related biomarkers, including hypothalamic-pituitary-adrenal (HPA) axis hormones and cytokines, given the implication of these biomarkers in stress-related pathologies (*Soria et al., 2018*; *Russell & Lightman, 2019*). Future studies also might want to address the study of faeces, as gut microbiota has emerged as a key player in the control of the HPA axis, especially during stressful situations caused by real or perceived homeostatic challenges (*Foster, Rinaman & Cryan, 2017*). Neuroimaging studies might also explore the neural correlates of toilet paper hoarding. Patients with hoarding disorder show higher dorsolateral prefrontal cortex (DLPFC) activation during tests of executive functions than do patients with OCD (*Hough et al., 2016*). OCD patients with prominent hoarding symptoms have also shown greater activation in the bilateral anterior ventromedial prefrontal cortex (VMPFC) than do patients without hoarding symptoms and healthy controls (*An et al., 2009*). As previous studies have shown dramatic improvement in anus wiping of an OCD patient after brain stimulation with tACS targeting the DLPFC, future studies might study the role of the prefrontal cortex in the pathogenesis of toilet paper hoarding.

## Study limitations

The main limitation of our study is the small number of studies included in our systematic review. A meta-analysis could not be performed for this reason, as in the protocol of our systematic review, we aimed to include a minimum of 5 studies with similar effect sizes for conducting a quantitative meta-analytical synthesis. We increased the number of publications with the realist review, and we also included grey literature, but the evidence generated from studies during the COVID-19 pandemic was particularly low. A negative publication bias on toilet paper hoarding is possible, as authors might avoid publishing articles dealing with toilet paper. Along these lines, negative outcomes associated with the pathological use of toilet paper (e.g., suicide cases secondary to toilet paper choking) might also be considered humiliating and be underreported in the scientific literature.

Finally, although we have proposed an algorithm for managing toilet paper hoarding or other pathological uses of toilet paper, it is important to emphasize that this algorithm has not been validated. Future studies might improve upon this limitation by testing and validating its application in clinical practice. If our algorithm is validated in future studies, it might be useful for psychiatrists and clinical psychologists who need to manage people with potential toilet paper hoarding behaviours.

Although our study has several limitations, it is also the first realist review exploring potential mechanisms that could explain in part the toilet paper hoarding experienced in many countries during the COVID-19 pandemic. Our study allows the identification of

gaps in the literature and will help researchers to design and conduct future studies aiming to better understand the causes and consequences of toilet paper hoarding in the general population and in people suffering from mental illnesses.

## CONCLUSIONS

The COVID-19 pandemic has been associated with a worldwide increase in hoarding behaviours, with toilet paper being one of the most desired objects. Social media and social cognitive biases seem to be major contributors to this hoarding behaviour and might explain some differences in toilet paper hoarding between countries. Other mental health-related factors are likely to be involved, such as the stressful situation of the COVID-19 pandemic, fear of contagion, or particular personality traits (conscientiousness). Future studies might help to better characterise the phenotype of toilet paper hoarding and to explore psychological and neurobiological mechanisms underlying this behaviour.

## ACKNOWLEDGEMENTS

This review is a proposal of the Catalan Toilet Tissue Research Group in Mental Health (CATOTIM), which is composed of psychiatrists and clinical psychologists interested in the study of the causes and consequences of pathological toilet paper use. All authors are members of the CATOTIM group. Current CATOTIM members are: Javier Labad, Alexandre González-Rodríguez, Jesús Cobo, Joaquím Puntí, Josep María Farré and Armand Guàrdia. As people were hoarding toilet paper amid the coronavirus pandemic, this study was driven by the interest in studying potential mechanisms linked to this behaviour that can cause distress to individuals.

### Funding

There was no specific funding for the current study. However, Javier Labad has received an Intensification for the Research Activity Grant from the Generalitat de Catalunya during 2018-2019 (SLT006/17/00012) and the Instituto de Salud Carlos III during 2020 (INT19/00071). Javier Labad, Alexandre González-Rodríguez and Jesús Cobo are researchers from the Research Group in Psychoneuroendocrinology and Stress in Psychosis, which has been funded by the AGAUR (2017SGR632). The funders had no role in study design, data collection and analysis, decision to publish, or preparation of the manuscript.

### Grant Disclosures

The following grant information was disclosed by the authors:
Generalitat de Catalunya during 2018-2019: SLT006/17/00012.
Instituto de Salud Carlos III: INT19/00071.
AGAUR: 2017SGR632.

## Competing Interests

None of the authors has received honoraria or collaborates with any toilet paper company.

## Author Contributions

- Javier Labad conceived and designed the experiments, performed the experiments, analyzed the data, prepared figures and/or tables, authored or reviewed drafts of the paper, and approved the final draft.
- Alexandre González-Rodríguez conceived and designed the experiments, performed the experiments, analyzed the data, prepared figures and/or tables, authored or reviewed drafts of the paper, and approved the final draft.
- Jesus Cobo conceived and designed the experiments, performed the experiments, analyzed the data, authored or reviewed drafts of the paper, and approved the final draft.
- Joaquim Puntí conceived and designed the experiments, authored or reviewed drafts of the paper, and approved the final draft.
- Josep Maria Farré conceived and designed the experiments, authored or reviewed drafts of the paper, and approved the final draft.

## Data Availability

All the data is available in the article and Supplemental Files.

## Supplemental Information

Supplemental information for this article can be found online at http://dx.doi.org/10.7717/peerj.10771#supplemental-information.

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
