# Peer review of "A systematic review and realist synthesis on toilet paper hoarding: COVID or not COVID, that is the question"

_PeerJ, doi:10.7717/peerj.10771_

## Round 0.1 · original submission · Minor Revisions

Thank you for your submission. As you can see the reviewers had some very different views on your paper, and my view is that this is an interesting and topical subject, therefore please address the concerns of ALL reviewers in making your revisions.

·

Basic reporting

The paper consists of three parts: A systematic review of studies on (determinants of) hoarding during the COVID-19 pandemic ("primary outcome"), a systematic review of inadequate use and hoarding of toilet paper in relation to mental health outcomes ("secondary outcome"), and a review of four predictors of hoarding that may be enhanced during the COVID-19 pandemic ("realist review"). Unfortunately, I am not convinced that these parts form a coherent whole. Moreover, I have severe doubts that enough data on the central topic are available to justify a systematic review.

Experimental design

A central problem is that the paper conflates hoarding as a clinically relevant mental health issue with 'hoarding' in the form of increased purchases in response to COVID-19. The authors discuss the use of clinical and non-clinical samples (l. 115-116). Yet, this does not address the implicit assumption that these two kinds of hoarding are the same or related constructs. Although I am no expert on clinical psychology, I find this highly doubtful. If a connection between these two constructs is made, it has to be supported and defended much more strongly.

Second, it is not quite clear which role toilet paper and the COVID-19 pandemic play in the reviews. The first part of the systematic review considers hoarding during COVID-19 (but not necessarily toilet paper). The second part considers inappropriate use of toilet paper (but not necessarily during COVID-19). Finally, the realist review sometimes focuses on toilet paper-specific outcomes (e.g., mechanism 1), but sometimes discusses hoarding more generally (e.g., mechanism 3). Ultimately, these three parts do not really hang together. There also exists little data which would allow for integration (e.g., there is no data providing evidence as to whether hoarding of toilet paper is in any way distinct from other hoarding behaviour, in either clinical or non-clinical populations). This suggests that the focus of the paper is too broad.

Validity of the findings

The first part of the systematic review examines studies on hoarding during COVID-19. At this point, data is extremely limited; the systemic review only highlighted two empirical studies. Importantly, preprints were included from the systematic review. The authors identify one preprint with empirical data (Columbus, 2020). However, there are several other preprints that report on purchasing behaviour during the COVID-19 pandemic (e.g., Bai et al., 2020; Bentall et al., 2020; Zettler et al., 2020). Given the limited available data, I find it problematic that preprints are excluded from the review. Even including preprints and generalising beyond toilet paper, however, there is scarcely enough literature on hoarding during the COVID-19 pandemic to justify a systematic review.

In contrast to the systematic review, which focuses on published literature, the realist review covers a wide range of publications. These not just include preprints, but also news articles and blog posts. Some of these make rather silly arguments (e.g., a Freudian analysis of "regression to the anal stage", l. 506-507---although I understood at least one of the cited authors to be using this as a toy example to explain Freudian reasoning of which they are critical). At times, the realist review is mostly essayistic in nature. For example, l. 426-439 discuss information diffusion on social media using the example of a toilet roll juggling challenge. It is not clear what to draw from such discussions, especially since the paper does not provide even a limited review of the actual empirical literature on the topic.

Additional comments

Minor issues
l. 260 describes the findings of Garbe et al. (2020) as 'excessive stockpiling'. However, the actual item simply referred to having more toilet paper than usual at home. This is plausibly described as stockpiling, but not excessive.

References
Bai, M. (2020). Who Bought All the Toilet Paper? Conspiracy Theorists Are More Likely to Stockpile During the COVID-19 Pandemic. Retrieved from https://psyarxiv.com/z2g34/
Bentall, R. P., Lloyd, A., Bennett, K., McKay, R., Mason, L., Murphy, J., … Shevlin, M. (2020). Pandemic buying: Testing a psychological model of over-purchasing and panic buying using data from the United Kingdom and the Republic of Ireland during the early phase of the COVID-19 pandemic. Retrieved from https://psyarxiv.com/u7vqp/
Columbus, S. (2020). Honesty-Humility, beliefs, and prosocial behaviour: A test on stockpiling during the COVID-19 pandemic. Retrieved from https://psyarxiv.com/8e62v/
Zettler, I., Schild, C., Lilleholt, L., Kroencke, L., Utesch, T., Moshagen, M., … Geukes, K. (2020). Individual differences in accepting personal restrictions to fight the COVID-19 pandemic: Results from a Danish adult sample. https://doi.org/10.31234/osf.io/pkm2a

Reviewer 2 ·

Basic reporting

There are some grammar mistakes, for example, “With no doubt, the COVID-19 has been the worse pandemic since the 1918 flu pandemic” should be “worst” instead of “worse”. Furthermore, the above sentence is also an example of ambigious reporting. What is meant by the COVID-19 pandemic being worse than the 1918 flu? Does it refer to the reproduction number, the economic impact or the rate at which it kills? Thus, language needs to be revised, both the grammar and accuracy of communication.

Most literature is adequately cited, however, the study is missing the consumer and retail service perspective. The literature on unusual purchasing has also studied toilet paper hoarding during COVID-19 (e.g. Laato, S., Islam, A. N., Farooq, A., & Dhir, A. (2020). Unusual purchasing behavior during the early stages of the COVID-19 pandemic: The stimulus-organism-response approach. Journal of Retailing and Consumer Services, 57, 102224.). Including research on this domain to the study would strengthen the paper. Perhaps first and foremost, these studies introduce the argument that toilet paper hoarding can in certain situations be rational behaviour.

The article structure is clear. Figures and tables are professionally drawn, informative and clear.

Experimental design

I like the idea that you looked into previous research on toilet paper hoarding, not just at the phenomena in the COVID-19 context. The methodology is adequately described with regards to the literature review and the realist review. The investigation is clearly reported and could be repeated by scholars. However, I have a few concerns regarding the experimental design and execution.

- The communication needs be improved with regards to reporting the research problem, hypotheses and research questions. Currently these appear in the Introduction section, but entangled with each other and the surrounding text. I am also confused by the choice of words, as you write “we aimed to conduct a systematic literature review…” instead of simply “we conducted a systematic review…”. Is conducting a SLR your research aim or rather the research method?

-In the materials and methods before jumping into the search string, you could explicitly mention that you are following the PRISMA systematic literature research method.

-The search string omits the important keyword “Toilet roll” (only looking at paper and tissue). I am not sure whether this would have had an impact on the discovered articles.

-the inclusion and exclusion criteria does not mention which types of documents you included (e.g. conference papers, journal articles, book chapters, editorial reports). Nor does it state whether you included only articles in English or articles in other languages as well.

-The results section does not state whether you screened the papers by title only, title and abstract or by assessing the full text articles. This could be clarified in the text even if it is visible in the PRISMA diagram (figure 1).

-The CATOTIM algorithm seems preliminary compared to the systematic review and the realist review. It seems odd that it is included in the materials & methods and the result sections. Still, the algorithm itself is well reported, interesting and worthy of discussion (Figure 3). To fix the issue, I would simply add it into discussion.

Validity of the findings

The findings are reported in an orderly fashion and are interesting. The reporting is well supported by Figures and Tables. However, the findings could benefit from a comparison to prior literature on hoarding

The one additional thing that concerns me is that in the CATOTIM algorithm and surrounding discussion there is no mention that hoarding toilet paper could under unusual circumstances be rational behaviour. That being said, overall, I found the discussion on the findings interesting and relevant.

Reviewer 3 ·

Basic reporting

Generally well-written and clear, with extensive relevant references and some historical background on key phrases.
One word I think misused - in the sections on "inadequate" use of toilet paper - which in colloquial English could be taken to mean "insufficient" use - which is not what the authors mean. I suggest revising to use the word "pathological" use of toilet paper or similar.

Experimental design

This is not original primary research - it is a systematic, empirically-based literature review.
But given the novelty of the topic and the fact that there's no structure in the rapidly growing literature that emerged in the past 6 months or so, it is an original systematic review of the literature.
Given that this is a survey - yes, the research question is welldefined, relevant and meaningful. The range of papers covered is detailed and wide-ranging so it is a rigorous review.

Validity of the findings

All underlying information about references is included. The conclusions are clear, well-stated and link with the aims and objectives of the paper.

---

## Round 0.2 · accepted · Accept

Thank you for your re-submission which we are pleased to accept for publication. Please address the issue with the Laato et al., 2020 reference while in Production.

Reviewer 2 ·

Basic reporting

As stated in my initial review, the paper is well written. With their revision, the authors managed to improve the structure and readability of the work even further.

One potential problem with the work is a mismatch between cited literature and the list of references. Going through the revisions I noticed that a study on unusual purchasing during COVID-19 (Laato et al., 2020) is cited but does not appear in the list of references.

Experimental design

I have no further comments on the experimental design.

Validity of the findings

I believe the authors have sufficiently addressed the concerns raised in my first revision.

Reviewer 3 ·

Basic reporting

Review of a revised submission - revisions address the commentary effectively

Experimental design

Review of a revised submission - revisions address the commentary effectively

Validity of the findings

Review of a revised submission - revisions address the commentary effectively

Additional comments

Review of a revised submission - your revisions have addressed the commentary effectively.